# Efficacy of dopamine, epinephrine and blood transfusion for treatment of fluid refractory shock in children with severe acute malnutrition or severe underweight and cholera or other dehydrating diarrhoeas: protocol for a randomised controlled clinical trial

Monira Sarmin,[1] Nusrat Jahan Shaly,[1] Tania Sultana,[1] Md. Tariqujjaman [ID],[1] Shamima Sharmin Shikha,[1] Nafisa Mariam,[1] Didarul Haque Jeorge,[1] Mosharrat Tabassum,[1] Baitun Nahar,[1] Farzana Afroze,[1] Lubaba Shahrin [ID],[1] Md. Iqbal Hossain,[1] Baharul Alam,[1] Abu Syed Golam Faruque [ID],[1] M Munirul Islam [ID],[1] Din-E-Mujahid Mohammad Faruqe Osmany,[2] Chaudhury Meshkat Ahmed,[2] Karim Manji [ID],[3] Niranjan Kissoon,[4] Mohammod Jobayer Chisti [ID],[1] Tahmeed Ahmed [ID][1]

For numbered affiliations see end of article.

**Correspondence to**
Dr Mohammod Jobayer Chisti;
chisti@icddrb.org

## ABSTRACT

**Introduction** Diarrhoea is one of the leading causes of under-5 childhood mortality and accounts for 8% of 5.4 million global under-5 deaths. In severely malnourished children, diarrhoea progresses to shock, where the risk of mortality is even higher. At icddr,b Dhaka Hospital, the fatality rate is as high as 69% in children with severe malnutrition and fluid refractory septic shock. To date, no study has evaluated systematically the effects of inotrope or vasopressor or blood transfusion in children with dehydrating diarrhoea (eg, in cholera) and severe acute malnutrition (SAM) or severe underweight who are in shock and unresponsive to WHO-recommended fluid therapy. To reduce the mortality of severely malnourished children presenting with diarrhoea and fluid refractory shock, we aim to compare the efficacy of blood transfusion, dopamine and epinephrine in fluid refractory shock in children who do not respond to WHO-recommended fluid resuscitation.

**Methods and analysis** In this randomised, three-arm, controlled, non-masked clinical trial in children 1–59 months old with SAM or severe underweight and fluid refractory shock, we will compare the efficacy of dopamine or epinephrine administration versus blood transfusion in children who failed to respond to WHO-recommended fluid resuscitation. The primary outcome variable is the case fatality rate. The effect of the intervention will be assessed by performing an intention-to-treat analysis. Recruitment and data collection began in July 2021 and are now ongoing. Results are expected by May 2023.

**Ethics and dissemination** This study has been approved by the icddr,b Institutional Review Board. We adhere to the 'Declaration of Helsinki' (2000), guidelines for Good

## STRENGTHS AND LIMITATIONS OF THIS STUDY

⇒ This study includes severely malnourished children with diarrhoea having fluid refractory shock.
⇒ This study evaluates the cardiac function of the study children at the bedside by point-of-care cardiac ultrasound.
⇒ This study is excluding children Down's syndrome or any chronic illness.
⇒ Evaluation of non-diarrhoeal children having shock is out of scope for this study.

Clinical Practice. Before enrolment, we collect signed informed consent from the parents or caregivers of the children. We will publish the results in a peer-reviewed journal and will arrange a dissemination seminar.

**Trial registration number** NCT04750070.

## INTRODUCTION
### Background and rationale

Diarrhoeal diseases are the second-leading cause of death in under-5 children and were responsible for the deaths of 370 000 children in 2019.[1] In Bangladesh, diarrhoea causes 7.3% of under-5 deaths in 2019.[2] Children with severe acute malnutrition (SAM) frequently present with diarrhoea, and fluid refractory shock is one of the common complications of diarrhoea.[3] The management of these children is challenging. There

is a lack of evidence-based guidelines for the management of fluid refractory shock in severely malnourished children. Optimised management of the altered haemodynamics in fluid refractory shock is yet to be explored and different opinions exist among experts, especially for malnourished children. Data from the Dhaka Hospital of icddr,b demonstrates that despite appropriate care, 69% of children having comorbidities such as severe malnutrition die from fluid refractory shock.[4]

The WHO recommends the administration of isotonic saline boluses in a dose of 15 mL/kg over 1 hour (maximum two times), followed by blood transfusion in a dose of 10 mL/kg over 3 hours in cases unresponsive to isotonic fluid boluses.[5] If blood pressure does not restore to the desired level after fluid bolus, the condition is termed as a fluid refractory/septic shock. Quick arrangement of safe blood is difficult, especially in resource-poor settings. The Fluid Expansion As Supportive Therapy (FEAST) trial showed significantly higher mortality associated with the administration of saline bolus or albumin[6] in children having severe sepsis. Inotropes such as dopamine perform better than excessive fluid administration[7] when myocardial dysfunction or vasoregulatory failure results in shock rather than absolute hypovolaemia.[7 8] Recent studies also demonstrated that epinephrine resulted in the early resolution of shock with survival benefits than dopamine.[9 10] It is a complex situation whether to use blood or inotropes in fluid refractory shock. Severely malnourished children are in reductive adaptation with different haemodynamics than well-nourished children. Despite the very high death rates, the management of fluid refractory shock in severely malnourished children with early inotropes or vasopressors has never been studied.

Malnourished children have less left ventricular mass[11 12] as well as left ventricular function as evidenced by echocardiography,[11 13] which may constitute a bad prognostic parameter. The FEAST trial reported that in the bolus group, a significantly higher proportion had a terminal event cardiac rather than respiratory or neurological.[14] Point-of-care (POC) cardiac ultrasound is an emerging tool to assess cardiac function and fluid responsiveness in shock and it is non-invasive. Using the POC ultrasound to differentiate hypovolaemic shock from distributive shock, for example, fluid refractory/septic shock, measuring ventricular function and inferior vena cava (IVC) collapsibility can reduce the diagnostic time and increase the accuracy of diagnosis.[15 16] Usually, in hypovolaemic shock, both end-diastolic and end-systolic volumes are reduced,[16] left ventricle (LV) is small having a hyperdynamic wall motion.[17] Whereas in fluid refractory/septic shock, LV is hyperdynamic[17] and only end-systolic volume is reduced.[16] Again, the diameter of the IVC at inspiration and expiration and their ratio provides useful information regarding fluid responsiveness.[16] To date, no study has evaluated systematically the effects of vasopressor and inotrope(s) in children with dehydrating diarrhoea (eg, in cholera) and SAM having fluid refractory shock

unresponsive to conventional WHO-recommended fluid therapy.

The proposed study will provide valuable data to determine whether resuscitation strategies including vasopressor and inotropes or blood result in increased survival in children who are severely malnourished with fluid refractory shock and diarrhoea. In addition, POC cardiac ultrasound will provide information on whether the shock resulted from a fluid deficit or cardiac dysfunction.

## Objective

The study aims to compare the efficacy of blood transfusion, dopamine and epinephrine in fluid refractory shock in children having SAM or severe underweight who do not respond to WHO-recommended fluid resuscitation.

## Trial design

This is a randomised, three-arm, controlled, non-masked clinical trial in children 1–59 months old with SAM or severely underweight and shock (figure 1). With parental written informed consent, children are allocated to the study interventions following randomisation, in addition to usual supportive care that includes broad-spectrum antibiotics, oxygen inhalation, intravenous glucose, etc. The study protocol follows the Standard Protocol Items: Recommendations for Interventional Trials guidance for protocol reporting (modified to reflect the ongoing study activities)[18] (table 1).

## METHODS
## Participants and interventions and outcomes
### Study setting

The study is being conducted at the Dhaka Hospital of the icddr,b, Dhaka, Bangladesh. icddr,b is the largest diarrhoeal diseases hospital in the world. Annually a total of 194838 patients of all ages and genders got treatment and care from icddr,b, and 57.2% were under 5 years of age. In addition to diarrhoeal illness, many patients have other health conditions, such as pneumonia, malnutrition, sepsis and electrolyte abnormalities. For critically ill patients, there is an intensive care unit (ICU) facility present within the hospital, equipped with contemporary life support measures, including mechanical ventilators, bubble continuous positive airway pressure (BCPAP) oxygen therapy, high-flow nasal cannula oxygen therapy and syringe pumps for vasopressor support. The ICU of the Dhaka Hospital of icddr,b manages around 1000 paediatric patients each year. More than 80% of them have a poor socioeconomic background. About one-third have severe malnutrition, and one-fourth of them have severe sepsis or fluid refractory shock, there are other different wards for the treatment of diarrhoeal illness, respiratory problems and malnutrition. icddr,b possesses a well-equipped laboratory service capable of performing a range of clinical tests proposed in this study.

### Inclusion criteria

Inclusion criteria are children aged 1–59 months of either sex having diarrhoea and SAM who develop fluid

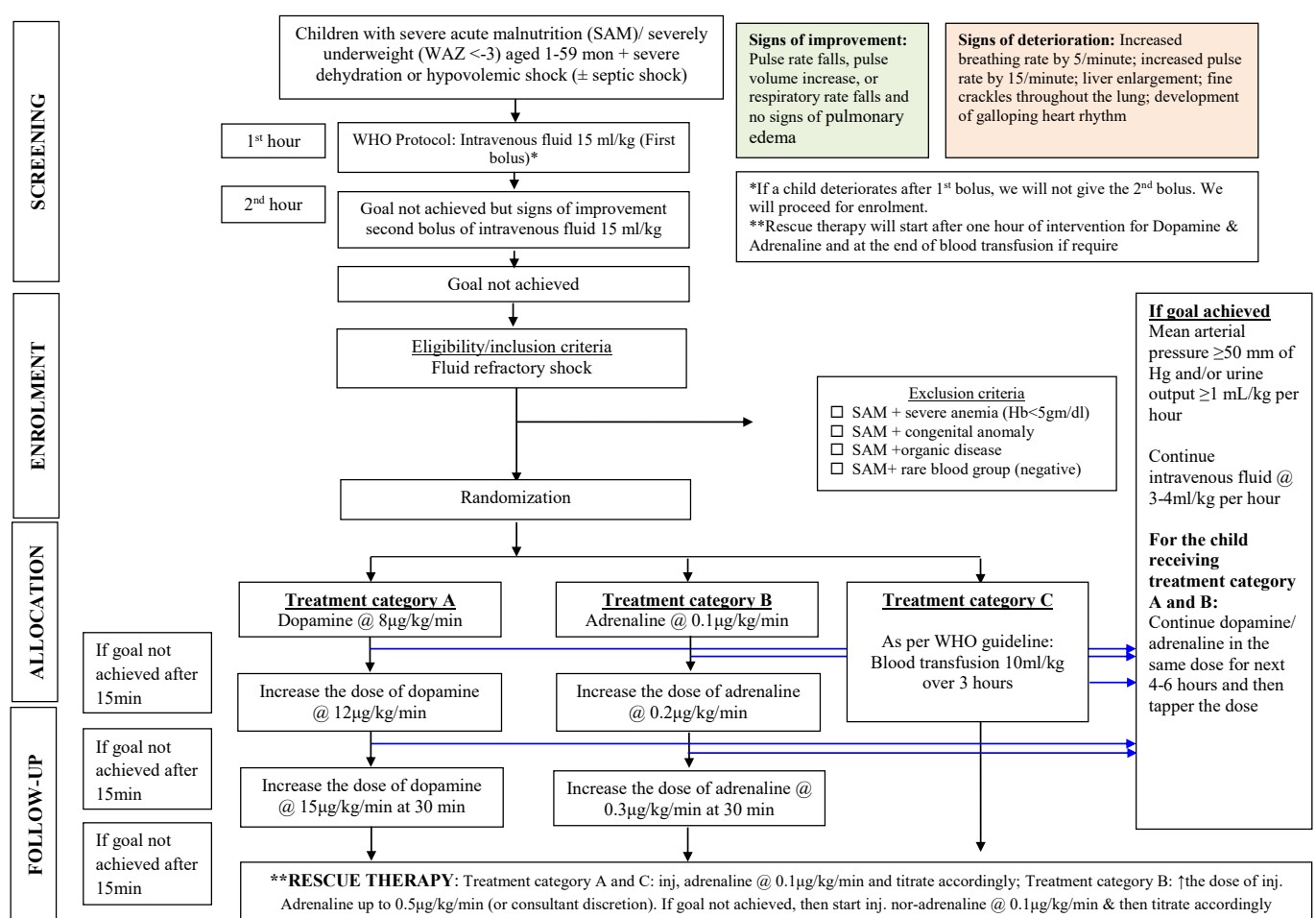

**Figure 1** Study flow chart showing the procedures of screening and enrolment and allocation of a child in any of the three treatment categories and ways of rescue therapy if the study goal is not achieved. Hb, haemoglobin: WAZ, weight for age.

refractory shock either at admission or at any point during their treatment in the hospital (which might be due to cholera or other dehydrating diarrhoea or severe sepsis). We also included children having severe underweight as they are also vulnerable and their mortality rate is also high. We are using corrected weight for the anticipated loss of fluids to calculate SAM or severe underweight.

### Exclusion criteria
Exclusion criteria are severe anaemia (haemoglobin (Hb) <5 g/dL) and sclerema where blood transfusion is a mandate, congenital anomalies (tetralogy of Fallot, TOF)/atrial or ventricular septal defect (trisomy 21, etc), negative blood groups, a requirement of cardiopulmonary resuscitation during screening and non-consent from the caregivers/parents.

### Randomisation
The permuted block randomisation technique is followed to select treatment arms for each child. The randomisation procedure was planned and executed by a scientist from icddr,b who is not involved in the study. The randomisation list containing the subject identification numbers (IDs) and the corresponding group allocation remained concealed. IDs are chronologically assigned

to each new study participant. After randomisation, an opaque envelope containing the name of the allocated intervention group is opened.

### Study duration
Twenty-four months.

### Collection of baseline information
Any SAM children with fluid refractory shock, either at admission or developed at some point during their treatment period, within the defined age group are screened for study eligibility criteria by the study physician. According to the inclusion and exclusion criteria, parents or attending caregivers of those eligible children are invited for providing consent for their children to be enrolled in the study. A signed written informed consent form by parents or caregivers gives data concerning the study, attainable advantages and risks, and voluntary nature of participation, as well as the right to cancel the child's participation in the study at any time (even after giving the initial consent) while not providing any reason; after this, children are enrolled by the study physician. One copy of the signed consent document was given to the caregiver of every participant and another copy was reserved for the study documentation. A pretested case

**Table 1** Standard Protocol Items: Recommendation for Interventional Trials (SPIRIT) Figure. The schedule of enrolment, interventions, and assessments.

| | | Study period | | | | Follow-up after discharge | | |
|---|---|---|---|---|---|---|---|---|
| **Indicators** | **Screening** | **Enrolment and allocation** | **Study drugs administration and F-U** | **Rescue therapy** | **Hosp course up to disposal** | **1st follow-up** | **2nd follow-up** | **28th follow-up (over phone)** |
| Time point | Day 0 | Hour 0 | Hour 0–3 (1 hour for dopamine and epinephrine, 3 hours for Blood) | Hour 1—disposal (for dopamine and epinephrine) Hour 3—disposal (for blood) | | Day 7±2 | Day 21±2 | Day 28±3 |
| Enrolment: | | | | | | | | |
| Eligibility screen | x | | | | | | | |
| Informed consent | x | x | | | | | | |
| Randomisation and allocation | | x | X | | | | | |
| Interventions: | | | | | | | | |
| Different drugs (dopamine or epinephrine or blood) | | | X | | | | | |
| Assessments: | | | | | | | | |
| Demographic | | x | | | | | | |
| Comorbidities | | x | | | | | | |
| Physical examination | x | x | X | x | x | x | x | – |
| Primary outcome: | | | | | | | | |
| Case fatality | | | | | | | | x |
| Secondary outcomes: | | | | | | | | |
| Treatment failure | | | X | | | | | |
| MAP stabilisation at 48 hours after enrolment | | | | x | x | | | |
| Mechanical ventilation | | | X | x | x | | | |
| Incidence of heart failure | | | X | x | x | | | |
| Duration of ICU stay | | | | | x | | | |
| Duration of total hospitalisation | | | | | x | | | |
| Time-to-achieve recovery from shock | | | X | x | x | | | |
| Ventricular function by cardiac USG | x | x | | x | x (If require) | | | |
| IVC collapsibility | x | x | | x | x (If require) | | | |
| *Detection of leptin from stored blood | | x | | | | | | |

*According to the addendum (ERC approval on 21 October 2021), at the end of the study, stored blood samples will be tested for leptin.
F-U, follow-up; ICU, intensive care unit; IVC, inferior vena cava; MAP, mean arterial pressure ; USG, ultrasonography.

record form is used to collect relevant medical history information, including the nature and duration of illness, medication for current illness and any history of significant past illness including pneumonia and diarrhoea. The form is also used to collect information on sociodemographic characteristics, such as age, sex, religion, gestational age, parental age with education and occupations, source of drinking water and sanitation, type of fuel use, cooking place and smoking history, average monthly income of the family, number of siblings, and number of rooms in the home.

Information is also collected about each child's feeding practice, which included their history of breastfeeding, formula feedings, or alternative complementary feedings, and also immunisation status. Data on clinical examination findings are recorded by the study physician and include pulse, respiratory rate, axillary temperature, anthropometric measurements (ie, height, weight,

mid-upper arm circumference (MUAC) for calculation of Z score), chest auscultation, oxygen saturation, presence or absence of chest wall indrawing, cyanosis and mental status (ie, normal, irritable or lethargic) and response (according to alert, response to voice or pain, unconscious (AVPU) scale). Each child's weight is measured by using an electronic weighing scale with a precision of 0.1 kg; length is measured using a standard-length board with a precision of 0.5 cm and MUAC is measured using a standard MUAC tape with a precision of 0.2 cm by the trained and experienced study nurse of Dhaka hospital. Fever is defined when the axillary temperature is 38°C or greater. Respiratory rate is counted for a full 60 s by exposing the trunk when the child is awake and calm; the presence of lower-chest-wall indrawing is noted at the same time. Oxygen saturation is measured using a portable pulse oximeter (Handheld Vital Signs Monitors-NT1D, Solaris Medical Technology, USA) with a probe on a finger or toe when the child breathes in room air. Oxygen saturation of <90% is defined as hypoxaemia, which is an indication for supplemental oxygen therapy by using BCPAP. Blood group and Rh typing are done at the bedside and we exclude the child having negative blood groups.

Bedside Hb (if not known) is checked by using HemoCue Hb 301 analyser (HemoCue AB, Sweden) during the screening period to exclude severely anaemic children requiring an urgent blood transfusion. The length of ICU stays, the total length of hospitalisation and time-to-achieve-recovery (time required for the restoration of mean arterial pressure (MAP) after the starting of the intervention for fluid refractory shock) are recorded. We are also keeping a record of stool and urine volume. Any deterioration or adverse events (ie, signs of heart failure or need for mechanical ventilation) are also documented. Both the study and outcome are recorded including treatment failure/success, hospital discharge, referrals and deaths or left against medical advice.

## Laboratory testing and sample handling

We are performing laboratory tests after the enrolment of a child in the study that includes blood for haemoglobin percentage, total and differential counts of white cell count (WCC), haematocrit, blood culture and antimicrobial susceptibility of pathogens, serum electrolytes and creatinine, calcium and magnesium, serum bilirubin, serum lactate, C reactive protein, procalcitonin and cardiac troponin I, stool (rectal swab for cholera, Salmonella, Shigella and Campylobacter) and urine culture, chest radiograph, and arterial or capillary blood gas analysis. We are also doing a test for the exclusion of dengue during the dengue prevalent period in the country. We are checking serum lactate at enrolment and 4–6 hours after enrolment. Our target is to send samples for laboratory tests within 30 min to 1 hour of enrolment. When a child is in shock, very often it is difficult to get an adequate amount of blood for proposed testing. Then we prioritise tests based on clinical relevance. We are doing the blood

grouping, screening and cross-matching while we are following a child for study eligibility. We are performing cardiac ultrasonography during screening, enrolment and 4–6 hours later after enrolment. As we are enrolling children 24×7, the study physicians are doing the bedside cardiac ultrasound and storing the images of specific preselected parameters. Later the images are reviewed by experienced cardiologists to get the measurements.

We are archiving blood samples and planning to check serum leptin at the end of the study.

## Study intervention and procedure

This is a three-arm trial where the intervention arms (dopamine and epinephrine) will be compared with the WHO standard intervention (blood transfusion). All study children stay at ICU for close monitoring.

During the screening, children receive intravenous isotonic fluid @ of 15 mL/kg over 1 hour (maximum two boluses, ie, up to 30 mL/kg) as per WHO standard of care. The choice of isotonic fluid is Ringer's lactate/Hartsol (normal saline if any suspicion of hyperkelaemia, acetate if severe dehydration is predominant). Dextrose is added if a child has hypoglycaemia with a random blood sugar <3 mmol/L. The study team members work to assess the eligibility and discuss with the parents/caregiver for possible enrolment. The regular ICU team helps in patient care. At the end of the screening, if the fluid refractory shock is diagnosed, then the study team proceed with written consent to avert the gap between consent to starting appropriate management of the child.

Children with fluid refractory shock are randomised to blood transfusion, dopamine or epinephrine therapy. Children in the dopamine arm (treatment category A) receive dopamine, 8 µg/kg/min (increasing the dose after 15 min to 12 µg/kg/min to a maximum of 15 µg/kg/min, up to the achievement of goal). Children in the epinephrine arm (treatment category B) receive epinephrine, 0.1 µg/kg/min (increasing the dose after 15 min to 0.2 µg/kg/min to a maximum of 0.3 µg/kg/min, up to the achievement of goal). We are using peripheral lines to give dopamine or epinephrine. Drugs are prepared as intravenous infusion and instilled by an infusion/syringe pump. Children in the blood transfusion arm (treatment category C) receive a transfusion of whole human blood at a dose of 10 mL/kg over 3 hours (figure 1). To arrange blood within the time frame, we have an institutional volunteer donor list, two medical technologists dedicatedly work for this study, and we are also excluding children having negative blood groups. Even after that, If blood could not be arranged within 1 hour after the event of fluid refractory shock, we will start rescue therapy considering the event as treatment failure. In addition to the study interventions, broad-spectrum antibiotic therapy, oxygen therapy and frequent monitoring, all children receive intravenous maintenance fluids with half-strength normal saline and 5% dextrose @ of 3 mL per kg per hour during the arrangement of blood transfusion and intervention with inotropes. Along

with maintenance fluid, we add intravenous fluid if diarrhoea continues or a child develops dehydration. Intravenous fluid maintenance is continued for 6 hours after the achievement of a goal and is further continued if the goal is not achieved. Our goal is to achieve MAP>50 mm Hg and/or urine output ≥1 mL/kg per hour. After randomisation, we monitor children at every 15 min interval during the intervention to assess mental status by using AVPU, an acronym from 'alert, verbal, pain, unresponsive', capillary refill time, rate and volume of the radial pulse, cold/warm extremities, urine output, blood pressure, respiratory rate, peripheral capillary $O_2$ saturation and features of heart failure. We measure BP manually using the standard cuff. To minimise the use of a catheter, we apply a paediatric urine collector bag and measure the urine output. However, for a female child, it is difficult to measure. We are considering an indwelling catheter if required. Mechanical ventilation is provided if the child did not respond to bubble CPAP (if SpO2<85% at least after an hour of bubble CPAP oxygen therapy) or had a respiratory failure (pH<7.2 and Partial pressure of carbon dioxide ($PCO_2$)>60 mm Hg in ABG analysis or Oxygen saturation to fraction of inspired oxygen ratio ($SpO_2/FiO_2$)<265). Children who develop complications such as acute abdomen not responding to conservative management or acute renal insufficiency requiring dialysis are referred to a suitable higher facility for proper management. At the achievement of the goal, we continue the dose of dopamine or epinephrine for 6 hours. We follow up with the child every 30 min to 1-hour interval. Thereafter, we reduce the dose in a stepwise manner similar to how we increased the dose. If the child becomes hypotensive again, we reintroduce the previous dose and follow the same procedure for further iteration.

### Rescue therapy
The failure of any intervention is declared as a treatment failure. Then for children in the dopamine and blood transfusion arm, we start epinephrine as rescue therapy in a stepwise manner starting from 0.1 µg/kg/min and increasing every 15 min by 0.1 µg/kg/min up to restoration of the desired goal (MAP>50 mm Hg or urine output >1 mL/kg/hour) and the highest dose is 0.5 µg/kg/min (or consultant discretion) For children in the epinephrine arm, the dose is increased every 15 min from 0.3 µg/kg/min (the highest proposed intervention dose) to a max of 0.5 µg/kg/min. On achieving the goal, we continue the same dose for 4–6 hours and then we de-escalate in a stepwise manner.

If the goal is not achieved with injection of epinephrine then injection of norepinephrine is started @ 0.1 µg/kg/min and we increase the dose every 15 min by 0.1 µg/kg up to restoration of the desired goal (MAP>50 mm Hg or urine output >1 mL/kg/hour) and the highest dose is 0.5 µg/kg/min. On achieving the goal, we continue the same dose for 6 hours and then will de-escalate in a stepwise manner. For those patients whose MAP is not restored with inotropes, injection hydrocortisone @ 2 mg/kg every

6 hours is added.[19] During rescue therapy, for doses and selection of any inotrope(s), we are following the recommendation from the Paediatric intensive care unit (PICU) consultant as all cases are not the same.

### Feeding and other care
Depending on the clinical condition either we keep the child NPO (nothing per oral) or start trophic feeding for children who had MAP>50 mm Hg and had no evidence of paralytic ileus. Feeds consist of 'Milk Suzi' (similar to WHO-recommended F-75) for children older than 6 months, and modified infant formula (in addition to breast milk, if applicable) is given to infants 1–6 months of age.[20] Thus, we provide a fluid volume of 120 mL/kg/day to deliver 80–82 kcal/kg/day. A child is kept warm to prevent hypothermia. If there is hypoglycaemia (random blood glucose <3 mmol/L), correction is given with 10% glucose intravenously in a dose of 5 mL/kg to be followed by 10% glucose administration through a nasogastric tube, following the standard protocol of the hospital.

### Nutritional rehabilitation
As the study children have SAM, they require nutritional rehabilitation. After recovering from acute illness, the mothers/caregivers are asked to stay in the hospital's nutritional rehabilitation unit (NRU) for 2–3 weeks. In the NRU, the standard dietary protocol is followed for nutritional rehabilitation and the children are discharged on fulfilling discharge criteria.

### Postdischarge follow-up
We follow up with a child on the 28th day (±3 days) of enrolment to know their clinical status over the phone. In addition, the children who were discharged from NRU or taking early discharge from the ward, are requested to come for a prescheduled follow-up at the hospital nutrition follow-up unit (NFU) of icddr,b. We followed the children at NFU and record their nutritional status and health problems. Our scheduled follow-ups for this study children are in two time points; once after 7 days of discharge, then after 15 days of first follow-up. If any mother/caregiver fails to attend a follow-up visit, then we communicate with the mother/caregiver over the phone for two time points. We document the outcome whether they are alive or not, and had any intercurrent illness, for example, diarrhoea, pneumonia.

### Outcome measures
Primary outcome variable is the case fatality rate (at 28±3 days from the day of enrolment). The secondary outcome variables are the rate of treatment failure, rate of MAP stabilisation at 48 hours after enrolment, need for mechanical ventilation, the incidence of heart failure, duration of ICU stay and total hospitalisation, time-to-achieve recovery, evaluation of cardiac function by POC ultrasonogram (right ventricular function, left ventricular function and IVC collapsibility).

## Sample size

The 2010–2011 data from the ICU of the Dhaka Hospital of the icddr,b (the proposed study site) revealed that among the 36 severely malnourished children with shock, 22 died, that is, mortality rate among SAM children with septic shock was 61% (22/36). We assumed that the proposed interventions would reduce the case fatality rate by 50%, that is, the rate will decrease from 61% to 31%.

To detect this magnitude of difference in the death rate, with 80% power (20% type II error) and 0.05 type I error, the required sample size is 40 children in each of the three groups. Considering around 10% drop-outs after enrolment, the total sample size would be 45 children at least in each of the three groups, that is, a total of 135 children—the final sample size of our study.

## Data collection, management and analysis

Data will be collected using a hard copy structured case record form. Data will be entered into a personal computer on the online platform named Research Electronic Data Capture. The data management team will check the consistency of data and will download and store the dataset as a backup to ensure data security. Statistical analyses will include descriptive as well as analytical methods. The effect of the intervention will be assessed by performing an intention-to-treat analysis as well as a per-protocol analysis. We anticipate that there will be no differences in the sociodemographic data among the three groups. Categorical variables will be compared using the $\chi^2$ test or Fisher's exact test as appropriate. When the variables of interest are continuous and distributed normally, the statistical significance of group means will be evaluated by analysis of variance. When the variables are continuous and not normally distributed, the statistical significance of group differences will be determined by the Kruskal-Wallis test. The post hoc test will be done if required. To determine the differences in primary outcome, that is, case fatality among groups, we will perform the $\chi^2$ test or Fisher's exact test as appropriate. The multiple generalised linear model (log-binomial regression) will be performed to explore the effect of interventions on case fatality after adjusting for potential confounders, including the child's age, child's sex, child's morbidity status, wealth status, etc. The strength of the differences will be determined by calculating the relative risks with 95% CIs. In addition, we will analyse the Kaplan-Meier survival curve to compare the survival times among the three treatment groups. A p<0.05 will be considered statistically significant. The scheduled interim analysis will be done after 50% of enrolment or earlier if the data safety and monitoring board (DSMB) advises. DSMB is an independent body, they review and follow the entire study and provide time-to-time suggestions. We will follow the stopping rule if the p<0.003, as recommended by Pato and Fleming.[21] Data analysis will be done using SPSS for Windows (V.20.0., IBM and Epi Info (V.7.0, USD, Stone Mountain, Georgia, USA) or STATA (V.15.0 SE).

## Reporting of adverse events

Serious adverse events (deaths) after initiation of the study intervention are recorded and presented to the DSMB within 24 hours of the event. The report includes a detailed clinical history, the study intervention and management given until the outcome. We contacted over the phone for study participants who were referred.

## Operational definition

### Severe acute malnutrition

A child with a <−3 Z-score of weight for height/length of the median value of the WHO standard or MUAC<115 mm (for children >6 months of age)/visible wasting (for children <6 months of age), or presence of nutritional oedema.

### Severe underweight

A child with a <−3 Z-score of weight for age of the median value of the WHO standard.

### Sepsis

Presence of infection/inflammation plus hyperthermia or hypothermia (temperature >38.5°C or<35.0°C, respectively) plus tachycardia (HR: infant >160/min, 1–5 years >140/min) or abnormal WCC (>12×10$^9$/L or, <4×10$^9$/L or, band and neutrophil ratio ≥0.1).

### Severe sepsis

Severe sepsis is defined as the presence of sepsis plus organ dysfunction or tissue hypoperfusion (poor peripheral perfusion (hypotension and/or absent peripheral pulses and/or delayed capillary refilling time (CRT) in the absence of dehydration or after correction of dehydration)).

### Fluid refractory shock or septic shock

Shock is defined as hypotension (MAP<50 mm Hg) and or reduced urine output <1 mL/kg/hour despite fluid resuscitation.[22] This 50 mm Hg cut-off was fixed on the basis of our local evidence from the same population.[22] A child in septic shock usually has a cold periphery, delayed CRT, low volume or absent pulse.

### Mean arterial pressure

MAP is calculated from systolic blood pressure (SBP) and diastolic blood pressure (DBP) by using a formula: DBP added to one-third of pulse pressure (subtraction of the DBP from the SBP).

### Treatment failure

Treatment failure is defined if an intervention results in
► MAP<50 mm Hg.
► Urine output is <1 mL/kg/hour, at the end of intended intervention over 2/3 hours.
  Heart failure (a combination of findings)
► Age-specific tachypnoea.
► Tachycardia.
► Enlarged tender liver.
► Pedal oedema (new or worsening).

- ► Basal crackles.
- ► Gallop.
- ► Response to furosemide.

Signs of improvement (during first hour/second hour of fluid bolus therapy).

- ► Pulse rate falls.
- ► Pulse volume improves.
- ► Respiratory rate falls.
- ► No signs of pulmonary oedema.

Signs of deterioration (during first hour/second hour of fluid bolus therapy).

- ► Breathing rate increases by 5/min.
- ► Pulse rate increases by 15/min.
- ► Liver enlarges (by measuring size at the beginning of fluid resuscitation and at the time of deterioration).
- ► Fine crackles at the lung base or throughout the lung fields.
- ► Oedema (new or worsening).
- ► Galloping heart rhythm.

### Acute kidney injury

Acute kidney injury is defined as an increase in serum creatinine by ≥0.3 mg/dL from baseline within 48 hours or, an increase in serum creatinine to ≥1.5 times baseline or age-specific upper limit of reference interval or, urine volume ≤0.5 mL/kg/hour for 6 hours.

### Recovery time

Time required for the restoration of MAP after starting the intervention for shock.

### Congenital heart diseases

Atrial septal defect, ventricular septal defect, TOF.

### Organic/other diseases

Down's syndrome, cerebral palsy.

### Patient and public involvement

We have planned to conduct public engagement activities like round table discussions and workshops at the end of the study.

### ETHICS

This study has been approved by the institutional review board (IRB) of the icddr,b (Protocol No. PR#20021, version 2.2, dated 21 July 2022). We are conducting the study in compliance with the 'Declaration of Helsinki' (2000), Guideline for Good Clinical Practice. Before enrolment, we collect signed informed consent from the parents or caregivers of the children. We are strictly maintaining privacy, anonymity and confidentiality of data or information identifying the patients or caregivers. Participants' medical information, particulars of treatment and laboratory test results will be kept confidential and under lock and key; only the research staff will have access to this information. The risk to enrolled subjects from participation in the study is minimal. Adverse medical events are anticipated as the study is enrolling critically ill patients. This protocol has been reviewed and approved by the DSMB of icddr,b. DSMB is a five-membered board that independently reviews the activity of the study. We are reporting the occurrence of any serious adverse event (eg, death) to the DSMB within 24 hours of the event and will follow their recommendations. There is no influence of the funder at any stage of the research, starting from study design and up to publication. Study patients are recruited into three different groups. For this clinical trial, injection epinephrine and injection dopamine are being used in fluid refractory shock. These drugs are already in use in the ICU of Dhaka hospital of icddr,b.[22] WHO recommends blood transfusion, and we are routinely practising it. Regardless of the study result, finally, it will be disseminated.

### Strategies for achieving adequate participant enrolment

Treatment at icddr,b Dhaka Hospital is free of cost for all patients. In addition to routine care, the study children get more follow-up by study physicians. So, they get the benefit of early detection and early intervention for any clinical condition. The caregiver has access to the study investigators and other physicians to communicate and to get an answer to their questions related to their child's medical condition and treatment even after their discharge. However, we pay a small amount of money (transport cost) for their follow-up visit after their usual discharge. We believe this information and empathetic care from the study will ensure adequate participant enrolment to reach the target sample size and also reduce the number of lost to follow-up.

### Dissemination

After approval from the investigators of the protocol, we will publish the results and other findings of the study. We will use The International Committee of Medical Journal Editors guidelines to ascertain the authorship of papers.

### Study status

As of August 2022, we have screened 49 children and enrolled 18 participants. Participants' screening and enrolment are ongoing.

### Protocol amendments

After the first approval of the protocol, we included two cardiologists as coinvestigators. After 1 year of enrolment, we made a change in the inclusion criteria also. Now we are including severely underweight children along with SAM children. So, we required amendments to the protocol which were all reviewed and approved by the icddr,b IRB. Now we are using version 2.2 of the protocol.

**Author affiliations**
[1]Nutrition and Clinical Services Division, International Centre for Diarrhoeal Disease Research,Bangladesh (icddr,b), Dhaka, Bangladesh
[2]Department of Cardiology, Bangabandhu Sheikh Mujib Medical University, Dhaka, Bangladesh
[3]Department of Pediatrics, Muhimbili University of Health and Allied Sciences, Dar-es-Salaam, Tanzania

[4]Department of Pediatrics, University of British Columbia, Vancouver, British Columbia, Canada

**Acknowledgements** icddr,b is grateful to the governments of Bangladesh, Canada, Sweden and the UK for providing core/unrestricted support.

**Contributors** The study concept and design were conceived by MS, BN, LS, FA, MIH, BA, ASGF, MMI, MFO, MAC, KM, NK, MJC and TA. NJS, TA, FA, LS, SSS, NM, DHJ, MT, MJC and MS are conducting patient screening, patient management and data collection. Data management is performed by MT, NJS, TA, FA, SSS, NM, DHJ, MT and MS . Analysis will be performed by MT, MS, MJC, LS and TA. MS, LS, NJS, TS, MT and MJC prepared the first draft of the manuscript. All authors reviewed, edited and critiqued the manuscript.

**Funding** This protocol is supported by Wellcome Trust, grant number—215691/Z/19/Z.

**Competing interests** None declared.

**Patient and public involvement** Patients and/or the public were involved in the design, or conduct, or reporting, or dissemination plans of this research. Refer to the Methods section for further details.

**Patient consent for publication** Consent obtained from parent(s)/guardian(s).

**Provenance and peer review** Not commissioned; externally peer reviewed.

**ORCID iDs**
Md. Tariqujjaman http://orcid.org/0000-0002-0172-9501
Lubaba Shahrin http://orcid.org/0000-0002-5676-3280
Abu Syed Golam Faruque http://orcid.org/0000-0001-8343-4653
M Munirul Islam http://orcid.org/0000-0002-8780-8760
Karim Manji http://orcid.org/0000-0002-7069-6408
Mohammod Jobayer Chisti http://orcid.org/0000-0001-9958-3071
Tahmeed Ahmed http://orcid.org/0000-0002-4607-7439

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
