## [Reviewer comments · BMJ Open]

ARTICLE DETAILS

TITLE (PROVISIONAL)	Efficacy of dopamine, adrenaline and blood transfusion for treatment of fluid refractory shock in children with severe acute malnutrition or severe underweight and cholera or other dehydrating diarrhoeas: protocol for a randomized controlled clinical trial
AUTHORS	Sarmin, Monira; Shaly, Nusrat; Sultana, Tania; Tariqujjaman, Md.; Shikha, Shamima; Mariam, Nafisa; Jeorge, Didarul; Tabassum, Mosharrat; Nahar, Baitun; Afroze, Farzana; Shahrin, Lubaba; Hossain, Md. Iqbal; Alam, Baharul; Faruque, ASG; Islam, M; Osmany, Mohammad; Chowdhury, Meshkat A.; Manji, Karim; Kisooson, Niranjan; Chisti, Md.; Ahmed, Tahmeed

VERSION 1 – REVIEW

REVIEWER	Calis, Job C. J. Emma Childrens Hosp, PICU
REVIEW RETURNED	21-Nov-2022

GENERAL COMMENTS	In general this is a well described and designed study investigating an important clinical question: what is the best treatment for children with severe malnutrition and fluid refractory shock. Several points would improve the description (or design) of the study: - The introduction does not stress the complexity of treating shock in malnourished children.- The shock definition used is not clearly described in the protocol; only septic shock is specified which I assume is the definition used in the study. This definition is different from the WHO definition, whilst the WHO treatment guideline is used. It may be justifiable to use a different definition, however then this should be motivated. Lastly if the definition of MAP<50 mm Hg is used, does this also apply to all age groups included in the study?- Is the standard treatment correct (ref 5, page 3 and page 9) – Is glucose not added? Some practical elements may be of importance but are not clearly described - How is the adrenalin/dopamine given (peripheral lines or central lines which may take more time)- How will Informed consent be done given the very short timelines <2 hours- Where will the intervention take place (ICU?) and will this apply to all arms? Practical questions concerning the endpoints:
---

	- Is the goal of the Rx with dopamine/adrenaline (page 10, line 25 and 30) the same in children of 1 month and children of 59 months? - Are all children catheterised to monitor urinary output? - Will fluid boluses not be given once inotropes are started (and RR is not responding), especially as cholera may require up to 100ml/kg fluid resuscitation in non-malnourished children. Is not being able to administer a BTF within the hour also considered treatment failure (page 10 line 33). Statistical questions -A targeted 50% reduction is a very big improvement. Is this based on any other study or data, if not the study should be considered a proof of concept rather than a definitive answer? -Was the sample size/power calculation based on the dopamine group and adrenaline group combined vs the control population? -Why was underweight added and how will this affect the analysis? -What is the expected duration of enrolment?
--	--

REVIEWER	Subhi, Rami MCRI, Paediatrics
REVIEW RETURNED	22-Nov-2022

GENERAL COMMENTS	Thank you for the opportunity to review this paper. A protocol for a 3 arm open-label trial (dopamine vs adrenaline vs blood transfusion) for children 1-5 years with severe acute malnutrition or severe underweight, AND fluid unresponsive shock, AND diarrhoea. I think on the whole the study is very important and well thought out. There are some areas that would benefit from some clarification: Design and definitions: 1- What is the definition of 'shock', and what is the definition of 'fluid unresponsive'? The inclusion criteria includes children with 'severe dehydration' OR 'hypovolemic shock' - does this mean some of the included children will not have shock? In which case, how are you judging if they are fluid responsive? 2- How is blood pressure measured, and why choose a standard definition of MAP>50 as goal, despite the fact that the expected blood pressure for a 1 year old is very different to that of a 5 year old? 3- How are the drugs given - peripherally or via a central line? 4- What is the standard of care? Sample size calculations are based on mortality proportion of 61%. Is this mortality rate based on children managed just with fluid and blood, or were these children also treated with inotropes/vasopressors? Because if the latter - ie mortality rate is 61% despite treatment with fluid, blood ino/vasopressors - why would we think that the interventions in this study could reduce mortality by 50%? 5- It sounds like 'heart failure' will be a clinical diagnosis rather than based on cardiac index/function on echo. And so it is not clear what the echo outcomes will be used for? Need to further specify. Also, echocardiogram interpretation introduces subjectivity - how will this be accounted for? How many researchers are performing and interpreting the echos? Are they blinded to treatment allocation? Statistics: - Many different methods are mentioned. Need to further specify the method particularly for primary outcome, and what confounders will be considered, and what sort of regression will be done
--

VERSION 1 – AUTHOR RESPONSE

Reviewer: 1

Dr. Job C. J. Calis, Emma Childrens Hosp, Kamuzu University of Health Sciences

Comments to the Author:

In general this is a well described and designed study investigating an important clinical question: what is the best treatment for children with severe malnutrition and fluid refractory shock.

Response: Thank you very much. We are working to find the best treatment option for children having severe acute malnutrition or severe underweight with fluid refractory shock and diarrhea.

Several points would improve the description (or design) of the study:

- The introduction does not stress the complexity of treating shock in malnourished children.

Response: Thank you. The treatment is complex as mentioned by the reviewer. This protocol is looking at the most effective management strategy either inotrope or blood in managing the shock, as you know fluid alone in severe malnutrition (with reductive adaptation) may be risky for precipitating cardiac failure. Now we are adding this in the introduction section. (page 3)

- The shock definition used is not clearly described in the protocol; only septic shock is specified which I assume is the definition used in the study. This definition is different from the WHO definition, whilst the WHO treatment guideline is used. It may be justifiable to use a different definition, however then this should be motivated. Lastly if the definition of MAP<50 mm Hg is used, does this also apply to all age groups included in the study?

Response: A child in shock usually has a cold periphery, delayed CRT, low volume or absent pulse. An objective parameter MAP is added to make the distinction clearer. If MAP remained <50 mmHg even after WHO recommended fluid bolus, we are describing it as a fluid refractory shock. we are using MAP <50mmHg as the cut-off point in these children. We used to follow this in our ICU and have published article using this cut-off. The median age of children with shock in our published was 5.3 months (IQR- 3.2, 12) (page 14), which is consistent in our ongoing recruitments. we have 3 treatment groups- Dopamine, adrenaline and blood. Blood is recommended by the WHO whereas surviving sepsis campaign guideline recommends adrenaline or dopamine. The only existing guidelines are that of WHO for severely malnourished children, which we shall request an update after the findings from this study.

- Is the standard treatment correct (ref 5, page 3 and page 9) – Is glucose not added?

Response: Thank you for your query. You are right. WHO recommends 5% dextrose with a fluid bolus. As we check and follow up blood sugar for every child, we do not add sugar with bolus fluid unless there is hypoglycaemia with a random blood sugar <3 m mol/l. (page 10) Glucose is added with maintenance fluid when a child is kept NPO for maintaining glycaemia. (page 10)

Some practical elements may be of importance but are not clearly described

- How is the adrenalin/dopamine given (peripheral lines or central lines which may take more time)

Response: Thank you for your concern. We are using peripheral lines to give dopamine or adrenaline. Drugs are prepared as an IV infusion in required doses and instilled by an infusion/syringe pump. (page 10)

- How will Informed consent be done given the very short timelines <2 hours

Response: Thank you for your concern. After a child is placed in ICU for management, the regular ICU team start the initial assessment and management. Simultaneously, the study team work to assess the eligibility and discuss with the parents/caregiver for possible enrolment. At the end of two hours, if a fluid refractory shock is diagnosed, then the study team proceed with written consent to

avert the gap between consent to starting appropriate management of the child. (page 10)

- Where will the intervention take place (ICU?) and will this apply to all arms?

Response: We are conducting this study at the ICU of icddr Dhaka Hospital. And this is applied to all three groups. (page 10)

Practical questions concerning the endpoints:

- Is the goal of the Rx with dopamine/adrenaline (page 10, line 25 and 30) the same in children of 1 month and children of 59 months?

Response: Thank you for the query. Yes, it is 1 month to 59 months. We are using MAP <50mm Hg as the cut-off point in these children and we have already provided local evidence for this cut-off.

- Are all children catheterised to monitor urinary output?

Response: Thank you. We do not catheterize all children. To minimize the use of a catheter, we apply a paediatric urine collector bag and measure the urine output. (page 11)

- Will fluid boluses not be given once inotropes are started (and RR is not responding), especially as cholera may require up to 100ml/kg fluid resuscitation in non-malnourished children.

Response: Thank you for your concern. Usually, we start inotrop after the fluid bolus or if the child deteriorates (in terms of increasing RR and HR) while the fluid bolus is ongoing (figure 1). While a child is on inotrope, we give maintenance fluid for ongoing diarrhoea. You are right, cholera may require up to 100ml/kg fluid resuscitation. However, after an infusion of 30ml/kg, we used to achieve our goal (MAP:50 mmHg) almost in all cases unless there is high purging (>15ml/kg/hr) or persistent vomiting (≥ 3 episodes/hr). For the two instances, we wait until the full replacement of outputs. Importantly, we observe occasional cholera among under-two children as our hospital surveillance showed cholera is <2% in this age group. (page 10)

Is not being able to administer a BTF within the hour also considered treatment failure (page 10 line 33).

Response: Thank you. We agree with you. Failure to arrange a blood transfusion within one hour after randomization will be considered as treatment failure. (page 10)

To avoid this, we have a donor list, two medical technologists dedicatedly work for this study, and we are also excluding children having negative blood groups.

Statistical questions

-A targeted 50% reduction is a very big improvement. Is this based on any other study or data, if not the study should be considered a proof of concept rather than a definitive answer?

Response: Thank you for your concern. We also agree 50% reduction is a big difference. We have some data (not published) from the same settings. Before the introduction of inotropes in 2007 and 2008, the mortality rate was very high, we followed WHO guidelines, and transfuse blood for cases who had a fluid refractory shock, however, we observed higher mortality among them. After the introduction of inotropes in 2009, our anecdotal observation found a significant reduction in mortality. From that point of view, we assumed a 50% relative reduction of death for our sample size calculation.

-Was the sample size/power calculation based on the dopamine group and adrenaline group combined vs the control population?

Response: The sample size calculation or power calculation was based on the combined groups and the minimum required sample was distributed for each of the three groups.

-Why was underweight added and how will this affect the analysis?

Response: Severely underweight children are also malnourished. They are vulnerable and their mortality rate is also high, hence excluding them makes the study results less generalizable. We believe that expanding the eligibility criteria in this way will strengthen our study and will widen our scope to enrol more participants to accomplish the study and this has been entrusted by our DSMB members. (pages 5 and 13)

-What is the expected duration of enrolment?

Response: Twenty-four months (page 6). We may assume we will require more time after observing the current trend of enrolment.

Reviewer: 2

Dr. Rami Subhi, MCRI

Comments to the Author:

Thank you for the opportunity to review this paper.

A protocol for a 3 arm open-label trial (dopamine vs adrenaline vs blood transfusion) for children 1-5 years with severe acute malnutrition or severe underweight, AND fluid unresponsive shock, AND diarrhoea.

Thank you very much for your kind review.

I think on the whole the study is very important and well thought out. There are some areas that would benefit from some clarification:

Design and definitions:

1- What is the definition of 'shock', and what is the definition of 'fluid unresponsive'? The inclusion criteria includes children with 'severe dehydration' OR 'hypovolemic shock' - does this mean some of the included children will not have shock? In which case, how are you judging if they are fluid responsive?

Response: Fluid unresponsiveness is defined as hypotension (mean arterial pressure <50mm of Hg) and or reduced urine output <1 mL/kg/h despite fluid resuscitation (page 15). A child in shock usually has a cold periphery, delayed CRT, low volume or absent pulse, mean arterial pressure <50mm of Hg and or reduced urine output <1 mL/kg/h. If a child has features of hypovolemic shock and there is a response to fluid therapy (improvement of blood pressure, heart rate and respiratory rate decrease after infusion) then they are not eligible for the study enrolment.

2- How is blood pressure measured, and why choose a standard definition of MAP>50 as goal, despite the fact that the expected blood pressure for a 1 year old is very different to that of a 5 year old?

Response: Thank you for your concern. We measure BP manually using a standard cuff. (page 11) We are using MAP <50mm Hg as the cut-off point in these children and we have local evidence (icddr, Dhaka hospital ICU) where we used this cut-off. Though our target age group is 1 month to 59 months, earlier we observed the median age was 5.3 months (IQR- 3.2, 12) from the study site (ref 22) and the age of our enrolled children in our ongoing study is also the same.

3- How are the drugs given - peripherally or via a central line?

Response: Thank you for your concern. We are using peripheral lines to give dopamine or adrenaline. Drugs are prepared as an IV infusion in required doses and instilled by an infusion/syringe pump. (page 10)

4- What is the standard of care? Sample size calculations are based on mortality proportion of 61%. Is this mortality rate based on children managed just with fluid and blood, or were these children also

treated with inotropes/vasopressors? Because if the latter - ie mortality rate is 61% despite treatment with fluid, blood ino/vasopressors - why would we think that the interventions in this study could reduce mortality by 50%?

Response: This 61% mortality was among the children who required blood transfusion as well as inotropes. However, the mortality among the shock patients responsive to fluid bolus was 15%. We have previously unpublished data from the same settings before the introduction of inotropes in 2007 and 2008 where patients unresponsive to fluid bolus and requiring blood transfusion hardly survive. After the introduction of inotropes in 2009, our anecdotal observation found a significant reduction in mortality. But no systematic trial was conducted to evaluate the efficacy including dopamine, adrenaline and blood.

5- It sounds like 'heart failure' will be a clinical diagnosis rather than based on cardiac index/function on echo. And so it is not clear what the echo outcomes will be used for? Need to further specify. Also, echocardiogram interpretation introduces subjectivity - how will this be accounted for? How many researchers are performing and interpreting the echos? Are they blinded to treatment allocation?

Response: Thank you. You are right. Heart failure is diagnosed clinically. As we have no cardiac function data from children having severe malnutrition and fluid refractory shock with diarrhoea, we believe these findings will help us to learn their baseline condition and to formulate some interventions in future.

As we are enrolling children 24X7, the study physicians are doing the bedside cardiac ultrasound and storing the images of specific pre-selected parameters. They were trained by qualified cardiologists who are part of the team also. Later the images are reviewed by the team to get the measurements. Six physicians are doing the cardiac ultrasound and two consultant cardiologists review the images later. The physicians are not blinded to the treatment allocation as it is an open-label trial. (page 10)

Statistics:

- Many different methods are mentioned. Need to further specify the method particularly for primary outcome, and what confounders will be considered, and what sort of regression will be done

Response: Thanks a lot for your comment. In the revised version, we have made changes and rearranged the sentences “in Data collection, management and analysis” section. According to your suggestion, we have added the types of analysis including multiple regression analysis as follows— “To explore the effect of interventions on case fatality, the multiple log-linear regression analyses will be performed after adjusting for potential confounders including child’s age, child’s sex, child’s morbidity status, wealth status etc. to reach more definitive conclusions.” (page 14)

VERSION 2 – REVIEW

REVIEWER	Calis, Job C. J. Emma Childrens Hosp, PICU
REVIEW RETURNED	16-Feb-2023

GENERAL COMMENTS	This is a well-designed study aiming at answering an important clinical problem. The study furthermore provide important data on the pathophysiology of shock. Nearly all questions have been answered (as far as I could check as I only received the revised document and no point by point response). The remaining ones may be handled by the editor. Minor remaining Issues - I would strongly suggest the use of urinary catheters to monitor urine output (one of two main selection criteria of the study and important parameter to monitor success/failure of treatment during the study), especially as patients have diarrhoea the use of
---

	collection bags are to me suboptimal/infeasible.  - What is the weight that will be used for determining if there is SAM or severe underweight. Is this the weight on admission (dehydrated weight) or the weight corrected for the anticipated loss of fluids? Ignorable issues for publication of this protocol (but possibly important for safety/ethics in general)  - The questions concerning power of the study are not addressed. - I am not used to adrenaline over a peripheral line, it can be done according to literature. I assume the authors have looked at safety issues such as a low concentration and regular checks of IV insertion sites? - The authors have now added that the MAINTENANCE fluid contains glucose, but it is not clear to me why the boluses in this study do not contain glucose (which is the suggested WHO treatment in malnourished children: If in shock or severe dehydration but cannot be rehydrated orally or by nasogastric tube, give IV fluids, either Ringer's lactate solution with 5% dextrose or half-strength Darrow's solution with 5% dextrose. If neither is available, 0.45% saline with 5% dextrose should be used (see Chart 8, p. 14, WHO pocketbook Pocket book of hospital care for children: guidelines for the management of common childhood illnesses – 2nd ed.)
--	---

REVIEWER	Subhi, Rami MCRI, Paediatrics
REVIEW RETURNED	07-Feb-2023

GENERAL COMMENTS	Thank you for your revisions and clarifications of the points previously raised. I have a few further comments, but on the whole this version is much clearer:  - Suggest being expanding in the introduction on definitions. An assumption is made that the included population has 'fluid refractory shock' / 'septic shock' (used interchangeably). However, the entry point for the study is malnutrition and dehydrating diarrhoeas. Presumably the reasoning is that if a child is refractory to initial fluid therapy, they do not have hypovolemic shock, and can be assumed to have 'septic shock'. This may be true for most children (without huge losses) but it is an assumption that should be explained more clearly - The sample size calculation is based on data that is more than 10 years old, and pre-inotropes, showing very high mortality rates. Management in this study is different in that children receive rescue therapy, and therefore mortality rate is expected to be lower than 61% in all groups. This is not a criticism of the design (it would be unethical to withhold inotropes), but I think in your interim analysis, the possibility that recruitment may need to be continued because there is no demonstrable difference in primary outcome should be considered - and therefore planned for in the protocol - In the section describing statistical analysis, please state the method for analysing the primary outcome (?multiple log linear regression)
---

VERSION 2 – AUTHOR RESPONSE

Reviewer: 2
Dr. Rami Subhi, MCRI

Comments to the Author:

Thank you for your revisions and clarifications of the points previously raised. I have a few further comments, but on the whole this version is much clearer:

Response: Thank you so much for the encouraging comments.

- Suggest being expanding in the introduction on definitions. An assumption is made that the included population has 'fluid refractory shock' / 'septic shock' (used interchangeably). However, the entry point for the study is malnutrition and dehydrating diarrhoeas. Presumably the reasoning is that if a child is refractory to initial fluid therapy, they do not have hypovolemic shock, and can be assumed to have 'septic shock'. This may be true for most children (without huge losses) but it is an assumption that should be explained more clearly

Response: Thank you so much for identifying the issue. Now we added it in the introduction section to make it understandable for the readers. (page 3)

- The sample size calculation is based on data that is more than 10 years old, and pre-inotropes, showing very high mortality rates. Management in this study is different in that children receive rescue therapy, and therefore mortality rate is expected to be lower than 61% in all groups. This is not a criticism of the design (it would be unethical to withhold inotropes), but I think in your interim analysis, the possibility that recruitment may need to be continued because there is no demonstrable difference in primary outcome should be considered - and therefore planned for in the protocol

Response: Thank you for your thoughtful suggestion. We are planning an interim analysis with a specific stopping rule after enrolling 50% of the study participants. We agree we may need to continue enrolment if there is no demonstrable difference in the primary outcome. However, for this, we will follow the direction of our data safety and monitoring board (DSMB). DSMB is an independent body, they review and follow the entire study and provide time-to-time suggestions. (page13)

- In the section describing statistical analysis, please state the method for analysing the primary outcome (?multiple log linear regression)

Response: Thank you for the suggestion. We will perform the Chi-square test or Fisher's exact test to determine the differences in case fatality (primary outcome) among groups. The multiple generalized linear models (log-binomial regression) will be performed to explore the effect of interventions on case fatality after adjusting for potential confounders including the child's age, child's sex, child's morbidity status. (page 13)

Reviewer: 1
Dr. Job C. J. Calis, Emma Childrens Hosp, Kamuzu University of Health Sciences

Comments to the Author:

This is a well-designed study aiming at answering an important clinical problem. The study furthermore provide important data on the pathophysiology of shock. Nearly all questions have been answered (as far as I could check as I only received the revised document and no point by point response). The remaining ones may be handled by the editor.

Response: Thank you so much for the positive comments.

Minor remaining Issues

- I would strongly suggest the use of urinary catheters to monitor urine output (one of two main selection criteria of the study and important parameter to monitor success/failure of treatment during

the study), especially as patients have diarrhoea the use of collection bags are to me suboptimal/infeasible.

Response: We agree with the reviewers. For a male child, it is easy to apply a pediatric urine collector bag. However, for a female child, it is difficult. So, we were considering an indwelling catheter as required. We are doing this procedure following standard aseptic precautions. (page 10)

- What is the weight that will be used for determining if there is SAM or severe underweight. Is this the weight on admission (dehydrated weight) or the weight corrected for the anticipated loss of fluids?

Response: Thank you. It is an important question.

We are using corrected weight for the anticipated loss of fluids to calculate severe acute malnutrition (SAM) or severe underweight. (page 5)

Ignorable issues for publication of this protocol (but possibly important for safety/ethics in general)

- The questions concerning power of the study are not addressed.

Response: Thank you. The sample size calculation or power calculation (80% power with 5% type 1 error) was based on the combined groups and the minimum required sample was distributed for each of the three groups.

- I am not used to adrenaline over a peripheral line, it can be done according to literature. I assume the authors have looked at safety issues such as a low concentration and regular checks of IV insertion sites?

Response: Thanks. Although the use of adrenaline in a peripheral line is tricky, we are using this through a peripheral line under routine vigilance of the intravenous site. There are dedicated ICU nurses for the study children.

- The authors have now added that the MAINTENANCE fluid contains glucose, but it is not clear to me why the boluses in this study do not contain glucose (which is the suggested WHO treatment in malnourished children: If in shock or severe dehydration but cannot be rehydrated orally or by nasogastric tube, give IV fluids, either Ringer's lactate solution with 5% dextrose or half-strength Darrow's solution with 5% dextrose. If neither is available, 0.45% saline with 5% dextrose should be used (see Chart 8, p. 14, WHO pocketbook Pocket book of hospital care for children: guidelines for the management of common childhood illnesses – 2nd ed.)

Response: Thank you. WHO recommends using fluid containing dextrose. However, as we are checking blood sugar at admission and thereafter four hourly or earlier depending on the clinical condition of the child, we are not adding glucose with the bolus fluid. However, if we find a child has low glucose, we consider bolus fluid containing dextrose.